# Time Trends of Tetrahydrocannabinol (THC) in a 2008–2021 German National Survey of Hemp Food Products

**DOI:** 10.3390/foods11030486

**Published:** 2022-02-08

**Authors:** Hannah Dräger, Ines Barthlott, Patricia Golombek, Stephan G. Walch, Dirk W. Lachenmeier

**Affiliations:** Chemisches und Veterinäruntersuchungsamt (CVUA) Karlsruhe, Weissenburger Straße 3, 76187 Karlsruhe, Germany; hannah.draeger@web.de (H.D.); ba.ines@gmx.de (I.B.); patricia.golombek@cvuaka.bwl.de (P.G.); stephan.walch@cvuaka.bwl.de (S.G.W.)

**Keywords:** hemp food, *Cannabis sativa* L., cannabinoids, tetrahydrocannabinol, food supplements, hemp tea, hemp seed, cannabidiol (CBD), time trends, national survey

## Abstract

∆^9^-Tetrahydrocannabinol (THC) is known as the main psychotropic compound present in the hemp plant. It also occurs in commercially available hemp food products and may have adverse effects on consumers. This article provides an overview of the current situation of the THC content in hemp food products in Germany in recent years. The content of THC was evaluated in a data set of 5 different hemp food product groups (tea, seeds, seed oils, food supplements, and nonalcoholic beverages) comprising 511 samples. For the toxicological assessment, the THC intake was estimated and the exhaustion of acute reference dose (ARfD) and lowest observed adverse effect level (LOAEL) was calculated using average daily consumption scenarios. Data show that hemp beverages and seeds typically do not contain amounts of THC that can exceed toxicological thresholds. On the contrary, hemp food supplements, such as cannabidiol (CBD) products, can contain high levels of THC, since the THC content of 18% of the samples has the potential to exceed the LOAEL and 8% even exceed the minimum intoxication dose. However, a significant linear decrease in the THC content of hemp food supplements was observed between 2018 and 2021 (*n* = 111, R = −0.36, *p* < 0.0001). A problematic food group is also tea based on flowers, leading to an increase in overall THC levels in recent years. Regulation of low-THC products within the framework of controlled distribution of cannabis for recreational use appears to be advisable.

## 1. Introduction

Recently, hemp products have experienced a strong increase in popularity. Increasingly more consumers are relying on the possible nutritional and health benefits of the hemp plant, especially the cannabinoids it contains. In addition to classical hemp-based food products (hemp seeds, hemp seed oil, hemp seed flour), a constantly growing product range includes pastries and pasta, candies, herbal teas, beverages, beers, cosmetic products, and food supplements [1,2]. In particular, products containing cannabidiol (CBD), e.g., “CBD oils”, are in high demand and comprise an increasing part of hemp products [2]. The term “CBD” originates from a non-psychotropic cannabinoid, which naturally occurs in the hemp plant. CBD has been researched for its various pharmacological effects and is approved as a medicinal product for the treatment of certain epileptic conditions [3]. Despite the lack of clinical evidence, CBD is advertised as a natural remedy for treating anxiety, depression, pain, inflammatory, and sleep disorders, and even cancer [4,5]. CBD-containing products are often advertised using unauthorized health or disease-related claims [1]. These products, e.g., CBD oils, are usually formulated with extracts of the leaves and flowers of the hemp plant *Cannabis sativa* L. Because CBD extraction processes for manufacturing so-called full-spectrum hemp extracts are not CBD selective, CBD products that are made from hemp extracts can contain a wide spectrum of cannabinoids naturally occurring in the plant.

Cannabinoids are plant-specific terpenophenolic C_21_ and C_22_ compounds, which occur naturally in the hemp plant [6]. The cannabinoids are predominantly formed in the glandular hairs, which are located on the flowers and leaves of the hemp plant. The roots and seeds of the hemp plant do not have glandular hairs and therefore initially do not contain cannabinoids [7]. Depending on which part of the plant the hemp-containing food products are made from, the products may contain widely varying contents of cannabinoids and other phytochemical components. In the hemp plant, about 120 different cannabinoids and cannabinoid acid analogs are described, out of which CBD and THC are the most abundant cannabinoids in terms of quantity [6]. Although CBD is not psychotropic, THC is known as the main psychotropic component of the hemp plant. Therefore, hemp is often used as a recreational drug in the form of marijuana or hashish. Psychotropic effects often include fatigue, dizziness, tachycardia, dry mouth, muscle relaxation, and increased appetite, among others [8]. The acute effects are perceived as pleasant and relaxing. However, the feeling of increased well-being may be replaced by dysphoria, anxiety, or panic attacks to several effects, such as perceptual changes, e.g., euphoria, hallucinations, slowed reflexes, and decreased coordination [9]. As a minimum intoxication dose, a value of 5 mg is assumed [10].

The legal regulations on hemp and its products are multilayered in Germany. While the hemp plant *Cannabis sativa* L. and the THC it contains, and the cannabis resin (hashish), are currently classified as non-marketable narcotics under the German Narcotics Act, hemp seeds, which generally contain none or only low levels of cannabinoids, are exempt from narcotics regulations unless they are intended for illicit cultivation [11]. Accordingly, hemp seeds and food products made from them, such as hemp seed oil or hemp seed flour, are not subject to the Narcotics Act [12]. Additionally exempt from narcotics legislation are plants and plant parts that originate from cultivation in European Union (EU) countries with certified seeds that do not exceed a THC content of 0.2% and whose use is also exclusively for commercial or scientific purposes that exclude abuse for intoxication [12]. The German government is currently planning to reform the Narcotics Act and legalize cannabis for recreational use.

According to current legislation, the cultivation of selected hemp varieties with low levels of THC has been permitted for food use in the EU [13]. However, not all products derived from fiber hemp are directly approved as food. According to the Novel Food Regulation (EU) 2015/2283, foods and food ingredients that have not been consumed by humans to a significant extent in the EU before 15 May 1997, are to be classified as novel and therefore require an approval [14]. With respect to the hemp plant *Cannabis sativa* L., according to the EU Novel Food catalog, extracts derived from the hemp plant and the cannabinoid-containing products made from them, and CBD, are classified as novel foods because the consumption of such products has not been demonstrated before 1997 [1,15]. According to this classification, CBD products are not authorized and therefore are not legally available as foods [1]. Cannabis leaves and flowers are also questionable as food business operators were unable to provide proof of consumption before 1997.

Regardless of the evaluation of the marketability of hemp-containing products due to their novel food status, the THC content of the product is decisive in determining the point at which a preparation can no longer be considered safe within the meaning of EU Regulation 178/2002. Despite the low THC content of the fiber hemp varieties approved for use in food, toxicologically questionable concentrations of THC may be present in fiber hemp products. To date, there are no mandatory maximum limits for THC in the EU.

In 2015, the European Food Safety Authority (EFSA) established a dose of 2.5 mg of THC per day (corresponding to 36 µg per kg of body weight in a 70 kg person) as the lowest observed adverse effect level (LOAEL) based on human data. From this, an acute reference dose (ARfD) of 1 μg THC per kg body weight was derived by adding safety factors [16]. ARfD is defined by the World Health Organization (WHO) as the amount of substance that can be ingested per kilogram of body weight in the diet with one meal or within one day without an identifiable risk. The percentage of ARfD exhaustion allows a potential health risk to be quantified and compared for a single consumption during a meal or in a day [17]. A degree of exhaustion of ARfD of more than 100% indicates that a possible risk can no longer be excluded with the required certainty.

In a recent statement, the German Federal Institute for Risk Assessment (BfR) recommends that the toxicological assessment of hemp-containing food products be conducted on the basis of the EFSA assessment [18]. If the ARfD of 1 μg/kg body weight THC is exceeded or if an intake of 2.5 mg THC/day is reached in general, hemp-containing foods are no longer considered safe and evaluated as either “unfit for human consumption” or “injurious to health” [10].

Because the THC content in hemp-containing food currently continues to be a major issue in the field of consumer protection, this article provides an overview of the current situation of the THC content and the exposure of consumers. The analysis is focused on the five main hemp food product groups (tea, seeds, seed oils, food supplements, and nonalcoholic beverages).

## 2. Materials and Methods

The samples of the present data were taken and examined by official food control laboratories in Germany in the context of governmental food control using validated and externally accredited methodologies according to ISO/IEC 17025. Afterward, the data were collected and consolidated by the German Federal Office of Consumer Protection and Food Safety (BVL) and published because of a consumer inquiry via the website FragDenStaat.de [19]. The consumer inquiry is based on the Consumer Information Act, which obliges food control authorities to provide consumers with information on food, feed, consumer products, and cosmetics [20]. The requirement to receive information includes, for example, unauthorized deviations from requirements, and measures and decisions taken in connection with the deviations, hazards, and risks to health and safety posed by products, monitoring activities, and measures for consumer protection [20]. The consumer inquiry relates to the increased reporting of foods with high THC content and the resulting warnings of undesirable health consequences due to the psychotropic effects associated with THC. Therefore, the requestor asked for information on the THC content of the products and the expected exposures. The original consumer inquiry including the full dataset can be accessed through the Internet portal FragDenStaat.de hosted by the nonprofit association Open Knowledge Foundation Deutschland e.V. (Berlin, Germany) [19].

The available data contain information about, e.g., the year and the type of shop of sampling, the country of origin, the product group and matrix of the sample, the THC content and the content of other cannabinoids, the analytical methods, and the limit of detection (LOD) and the limit of quantification (LOQ) of the applied method. In total, this dataset contains 1366 measured values for cannabinoids in various hemp food products. For the subsequent evaluation, only the values for the cannabinoid THC of five major food product groups were taken into account. Additionally, values for total-THC, which is a sum of the free form and its non-psychotropic acid, were excluded as they would overestimate exposure by about 50% and are unsuitable for risk assessment as they overestimate the psychotropic effects, specifically of products unintended for heating (which might lead to decarboxylation of the acid form of THC). In relation to the total provided values, this corresponds to a percentage of 37% that was finally included in this study (511 samples).

The samples were examined from 2008 to September 2021. Although the data used for the evaluation contain samples from 2008 for hemp beverages, from 2009 for hemp tea, from 2010 for hemp seed oil, and from 2012 for hemp seeds and hemp food supplements, most samples were examined in 2016, 2018, 2019, 2020, and 2021.

Most of the samples were taken from German retailers. In addition, some samples were obtained from German drugstores, health food stores, bakeries, producers, and Internet retailers.

The main country of origin of the samples is Germany (67%). Some samples were produced in Austria (8%) or Switzerland (3%) while the rest of the samples (22%) originated from countries, such as France, Czech Republic, Belgium, and China, or had an unknown origin.

The analytical methods used for the determination of the THC content in hemp food products were gas chromatography (GC) in combination with mass spectrometric (MS) methods and high-performance liquid chromatography (HPLC)-MS/MS methods. In the case of hemp seed oils, the method 13.04.19-1 according to paragraph #64 of the German Food and Feed Law (LFGB) for the determination of ∆^9^-tetrahydrocannabinol in hemp seed oil is available. The method 47.00-9 according to paragraph #64 of the LFGB can be used to determine ∆^9^-tetrahydrocannabinol in hemp-based tea-like products. Since high injector and column temperatures cause decarboxylation of cannabinoid acids, with GC-MS methods, only the determination of the sum of ∆^9^-tetrahydrocannabinol and ∆^9^-tetrahydrocannabinolic acid (total-THC) is possible, if no derivatization is performed [21,22,23]. Using HPLC-MS/MS methods, due to the lower temperature, the specific content of ∆^9^-tetrahydrocannabinol can be determined separately from its acid, [24] and this method is currently preferred for exposure assessment [1].

Data were evaluated as described in the following: The food products were summarized according to the corresponding product groups (hemp tea, hemp seeds, hemp seed oils, hemp food supplements, and hemp nonalcoholic beverages). Table 1 shows an overview of the food products that are included in the five different product groups.

Data were evaluated on the basis of the following upper bound assumption: If the concentration of the sample was lower than the LOD, the LOD of the corresponding method was used for further calculations and evaluations. If the concentration of the sample was lower than the LOQ, the LOQ of the method was used.

The THC content (mg THC/kg product) was converted to daily intake (mg THC/d) using the mean daily consumption (g food product/d). The corresponding daily consumption of the product groups is shown in Table 2. Since there are no data available on the daily consumption of hemp products, the daily consumption of the corresponding hemp free food products is used for further calculations and evaluations. For hemp beverages and hemp tea, the data of the German national consumption study (Nationale Verzehrsstudie II, NVS II) of the Max Rubner Institute (MRI) were used. The daily consumption of hemp beverages of 176 g/d accounts for lemonade, fruit drinks, and others, such as nonalcoholic beers [25]. For hemp tea, the daily consumption of herbal tea and fruit tea of 233.5 mL/d was used for evaluation [25]. This is equivalent to 2.335 g tea/d for a preparation of 2 g tea per cup (200 mL). According to the assumption of BfR in its statement, for the calculations, a 100% transition of THC to tea infusion was assumed [23]. Since the mean daily consumption of nuts and seeds is 2 g/d according to NVS II, this daily consumption was used for further calculations of hemp seeds [25]. For hemp seed oil, the daily consumption was set at 24.5 g/d according to the mean daily consumption of fats and oils of NVS II [25]. In the case of hemp food supplements, no consumption data have yet been published as part of an official consumption study. Therefore, an average consumption of 5 g/d for common hemp food supplements (average of 181 samples) was used for further evaluation [10].

The daily THC intake was compared to the ARfD, the LOAEL, and the minimum intoxication dose of THC. Afterward, the exhaustion of these toxicological assessment values was calculated. Furthermore, the mean, median, maximum, 90th, and 95th percentile were calculated (see Table 2). The THC contents of all single samples of the five different food product groups and the calculations can be found in the Appendix A. All statistical calculations described above were performed using Microsoft Excel version 2016 (Microsoft, Redmond, WA, USA). Time trend analysis of the THC content in hemp food products was created using OriginPro version 9.75 (OriginLab Corporation, Northampton, MA, USA).

## 3. Results and Discussion

### 3.1. THC Content in Hemp Food Products

#### 3.1.1. Hemp Beverages (Except Tea)

Figure 1 shows that the daily intake of THC from hemp-containing beverages is low compared to the other investigated product groups. In the calculation of the median, mean, and maximum daily THC intake from hemp-containing beverages, 61 samples from the years 2008 to 2020 were included. Of the 61 samples, the THC content of 7 samples (11%) was below the LOQ and 35 samples (57%) below the LOD. Thus, only one-third (31%) of the samples showed a quantifiable THC content.

It can be observed that the calculated mean THC intake (0.0002 mg/d) exhausts the ARfD with 0.3% to a low extent. The exhaustion of the ARfD by the mean THC intake (0.005 mg/d) is higher but still low with 7%. For the sample with the highest THC content (0.529 mg/L), the THC intake is 0.093 mg/d. This results in an ARfD exhaustion of 133% (see Figure 1).

An exceedance of toxicological assessment values, such as ARfD, depends on the daily consumption of the respective products but also, in particular, the THC content of the product. In the case of hemp-containing beverages, the amount of hemp in the product is therefore a decisive factor. A review of the ingredient lists of various hemp-containing soft drinks shows that in hemp-containing soda, ice teas, and energy drinks, the characteristic ingredient hemp is often listed on the ingredients list as hemp extract or hemp infusion, consisting of water and low-cannabinoid hemp leaves. The quantities according to the ingredient list of the products range from 0.25% to 15% and thus show high variance. Additionally, hemp was found in some products exclusively as a flavoring, e.g., cannabis aroma.

From this information, it can be deduced that the hemp ingredient in hemp-containing beverages often represents only a small proportion of the total product. Thus, the THC content possibly contained in hemp is diluted by other ingredients (e.g., water) contained in larger mass proportions, so that the toxicological assessment value in the products may not be reached. Additionally, for no other product category, the proportion of samples with an unquantifiable THC content was as high as for beverages. These observations show that the THC levels in beverages depend on the formulation of the product. In addition to the form in which hemp is used in the product (hemp infusion or flavoring), the amount of hemp ingredient characterizing the product and the hemp variety from which the ingredient was obtained and its solubility (see Section 3.1.4) also play a role. These factors can influence the THC content of the product and lead to an exceedance of the toxicological threshold value, resulting in the corresponding complaints about the products.

#### 3.1.2. Hemp Seeds

Figure 1 shows that the daily intake of THC from hemp seeds is on average below the ARfD. In the product group seeds, the THC content for the 93 samples from 2012 to 2021 was included in the calculations. Of the 93 samples, 89 samples (96%) showed a quantifiable THC content. Both the median THC intake of 0.001 mg/d and the mean THC intake of 0.008 mg/d are significantly below ARfD, with an exhaustion of 1% and 12%, respectively. For the sample with the highest THC content (196 mg/kg), the THC intake amounts to 0.392 mg/d. This results in an ARfD exhaustion of 560% (see Figure 1) whereas the exhaustion of the LOAEL is only 16%.

Taking into account the natural occurrence of cannabinoids in the hemp plant, it is common for products made from hemp seeds to contain only low levels of cannabinoids. This is because hemp seeds do not possess any cannabinoid-secreting glandular hair. The THC found in the analysis of hemp seed products is often the result of contamination of the seeds during harvest through contact with flowers [7], which can explain the exceptionally high value in some outlier samples.

#### 3.1.3. Hemp Seed Oil

Hemp seed oil is a vegetable oil obtained from the seeds of the hemp plant and is used as an edible oil [23]. Figure 1 shows that the daily intake of THC from hemp seed oils for the median calculation is low and below ARfD. Hemp seed oils are also products of hemp seeds and thus they consist of the cannabinoid-poor part of the hemp plant. Here, the analytical detection of THC is also due to possible contamination of the seeds with resin containing cannabinoids from the glandular hairs of flowers and leaves. In this product group, the THC content for 137 samples from the years 2010 to 2021 was included in the calculations. Out of 137 samples, 102 samples (74%) showed a quantifiable THC content.

Both the median THC intake of 0.049 mg/d and the mean THC intake of 0.153 mg/d fall significantly below the LOAEL, with exhaustion of the LOAEL of 2% and 6%, respectively. Whereas the median THC intake does not exceed the ARfD with an exhaustion of 70%, the mean THC intake does exceed the ARfD significantly, with an exhaustion of 219%. From the assumed daily consumption, a THC intake of 5.7 mg/d results in the maximum determined THC content (232.8 mg/kg). This intake of THC leads to a clear exceedance of LOAEL with an exhaustion of 226% and also exceeds the minimum psychotropic level.

#### 3.1.4. Hemp Tea (Based on Flowers and Leaves)

In contrast to hemp seeds and hemp seed oils obtained from the cannabinoid-poor parts of the hemp plant, the cannabinoid-rich leaves and flowers of the hemp plant are generally used in products, such as hemp tea and food supplements. Due to the use of the cannabinoid-rich plant parts, these product groups can contain high levels of THC. The THC content depends mainly on the THC content of the hemp variety used for the product. In the EU, only hemp varieties with less than 0.2% THC, called fiber hemp, may be used [1]. Especially because such products are consumed directly as tea infusion or food supplements consisting of concentrated plant extracts, the toxicological threshold values derived by the EFSA are significantly exceeded more often with these products.

As tea, dried flowers and leaves of hemp varieties with low THC contents are commonly consumed directly as an infusion [26]. In this product group, the results of 89 tea samples from 2009 to 2021 were available. Of these, 78 samples (88%) had a quantifiable THC content. The median THC intake from hemp tea is 0.049 mg/d, which is below ARfD and shows an ARfD exhaustion of 70%. The mean THC intake of 0.157 mg/d shows an ARfD exhaustion of 224% and is already clearly above the ARfD. The exhaustion of the LOAEL with 6% is still low.

However, the worst case scenario shows that consumption of the sample with the maximum THC content (588 mg/kg) may result in an estimated THC intake of 1.37 mg/d. Ingestion of such high levels exceeds the ARfD by almost 20-fold (ARfD exhaustion = 1961%). The LOAEL is thus exhausted to 55%.

These findings assume that 100% of the THC is transferred to the aqueous tea infusion. There is still controversy in the literature about this question. On the one hand, it can be expected that the transition rate of THC into the tea infusions may be limited due to the lipophilic character of THC and its low solubility in water [23]. On the other hand, other components in the tea matrix may act as a solubilizer. Experimental data from Knezevic et al. and Hazekamp et al. showed that the THC content in tea infusions is subject to fluctuations influenced by the water temperature, water volumes, and steeping and boiling time during the preparation of the tea infusion [26,27]. Knezevic et al. showed a decrease in the THC content of about 49% in the plant material of fiber hemp after tea infusion at 43 °C. Whereas, at 97 °C, an increase in the THC content of about 53% in the plant material residue was determined [26]. This effect can be due to the decarboxylation of tetrahydrocannabinolic acid (THCA) at higher temperatures [26]. However, they do not describe how much of the decarboxylated THCA was transferred to the tea infusion. Hazekamp et al. found that relatively more THCA was dissolved in boiling water than THC, probably due to the higher solubility of THCA compared to THC [27]. They also showed that even when the amount of THC content (about 174 mg, as the sum of THC and THCA) in 1 g of cannabis was potentially very high, the corresponding tea infusion contained only a fraction of THC (10 mg/L) [27]. According to them, the relatively low THC content is probably a result of the saturation of the water phase with THC [27]. However, this finding was obtained for THC-rich drug hemp, which is possibly not directly transferrable to fiber hemp. The BfR has recently reviewed the evidence and concluded that the assumption of 100% carryover is justified, as experimental data on the carryover point to high fluctuations [23].

#### 3.1.5. Food Supplements

In the product category of food supplements, the THC content of 131 samples from the years 2012 to 2021 was available. Of the 131 samples, 98 samples (75%) showed a quantifiable THC content. Hemp-containing food supplements comprise various products. In addition to omega-3-rich capsules with hemp seed oil, CBD products, for example, CBD oils and other products with hemp extracts in high doses, account for a major part of food supplements. Due to the use of full-spectrum hemp extracts in food supplements, these products usually contain, in addition to CBD, other cannabinoids and additional phytochemical components [28]. Based on the above-mentioned aspects, it is obvious that during the extraction of the hemp plant, beside CBD, other cannabinoids, such as psychotropic THC, may accumulate in hemp extracts. Therefore, it is also feasible that the highest THC content of all product groups was found in food supplements.

The median THC intake of 0.130 mg/d leads to an ARfD exhaustion of 186% and an LOAEL exhaustion of 5%. The mean value of the daily THC intake of 1.3 mg/d is already 10-fold above the median THC intake and exhausted the ARfD by about 20-fold and the exhaustion of the LOAEL is 54%.

Based on the highest determined THC content, the results show a maximum daily THC intake of 17 mg/d, which corresponds to an exceedance of ARfD by more than 250-times and LOAEL by more than 6 times (exhaustion of LOAEL = 675%). This shows that the consumption of certain food supplements could lead to an excess of the minimum intoxication dose of 5 mg [10].

### 3.2. Time Trend Analyses

Figure 2 shows the mean daily THC intake in recent years until the first half of 2021. The numbers of samples for every year and every product group are shown in Table 3. Due to the small number of samples for the corresponding product groups in 2017, data from this year are excluded from Figure 2 and Table 3. As shown in Figure 2, the mean daily intake of THC between 2016 and 2021 through the consumption of hemp seeds, hemp seed oils, and hemp tea is considerably low. The ARfD for THC is not exceeded by the consumption of hemp seeds at any time. Taking into account the mean daily THC intake through hemp seed oils, the ARfD is exceeded every year except 2019. The highest mean daily THC intake through hemp seed oil was observed in 2016, with an exhaustion of the ARfD of 523% and the LOAEL of 15%. Although the mean daily THC intake through hemp tea exceeds ARfD in every year between 2018 and 2021 with an exhaustion of 142% (2021) to 305% (2019) of ARfD, the LOAEL is only exhausted in 4% (2021) to 9% (2019).

Compared to the mean daily intake of THC through the 4 food product groups mentioned, hemp food supplements account for the highest proportion in each year between 2016 and 2021. Taking into account the mean daily THC intake through the consumption of hemp food supplements, the ARfD is exceeded in each year of the mentioned period. LOAEL is exceeded by the mean daily intake of THC through hemp food supplements in 2018 (exhaustion 133%) and 2019 (exhaustion 107%). Furthermore, the exhaustion of LOAEL in other years (18% in 2016, 35% in 2020, and 13% in 2021) is higher than for the other food product groups shown.

Regarding the results between 2018 and 2021, a significant decrease in the THC concentrations of food supplements (*n* = 111, R = −0.36, *p* < 0.0001) was observed using linear regression. This may be caused by improvements in the extraction techniques of the CBD oil industry. Furthermore, the increased focus of state authorities on the control of manufacturers and the increased awareness of producers of legal regulations and the consequences of violating the law could be reasons for the decrease in the mean THC content of hemp food supplements within the last four years.

However, for hemp tea (*n* = 71, R = −0.15, *p* = 0.22), no decrease in THC concentrations was observed from 2018 to 2021 with respect to the 5% significance level.

Figure 3 shows the evolution of the THC content in hemp food products during the period 1996–2021. A total of 577 hemp food products were included in the analysis while hemp food supplements were excluded from the time trend analysis. The analysis was compounded using our own previously published data [1,2] so that in this case, a time period back to 1996 was covered.

Overall, in the trend analysis, two periods can be observed. From 1996 to 2004, the graph shows a clear decrease in the THC content in hemp food products. However, from the 2010s until now, a significant increase in the THC content in hemp food products can be observed. Regarding all THC measurements between 1996 and 2021, a significant decrease followed by a significant increase (*n* = 511, *p* < 0.0001) was observed using a quadratic regression model. A limitation of this evaluation is the use of data for total-THC in the period before 2008. However, this limitation does not influence the trends observed in the two periods.

In our opinion, the very high levels after the legalization of fiber hemp in 1996 can be explained by the strong contamination of the hemp seeds with cannabinoid-containing resin. In 1997, for instance, cases of THC poisoning were reported by Meier et al. [29]. After eating salad prepared with hemp seed oil with a high THC content, four adult patients suffered from gastrointestinal disorders and psychological effects, such as tachycardia, conjunctival irritation, “high sensation”, and dysphoria [29]. After that, the situation improved significantly in the following years through quality controls or official controls. Since 2016, novel tea products have started to enter the market and led to another significant increase (which peaked around 2016 and remains at a high level). Especially due to the use of the flowers in such products, they contain very high contents of THC while previously only leaves were used in hemp teas.

### 3.3. Risk Assessment for Oil, Tea, and Food Supplements

Figure 4 shows the percent of the samples that exceeded the ARfD and LOAEL in recent years until the first half of 2021. The number of samples for every year and every product group are shown in Table 3.

As shown in Figure 4, regarding the daily intake of THC through the consumption of hemp seed oil samples, 71%, 42%, and 27% of the samples exceeded ARfD in 2016, 2018, and 2019, respectively, showing a decrease. On the other hand, in 2020, 42% of the hemp seed oil samples exceeded the ARfD. The percentage of 17% of the samples, which exceeded ARfD in 2021, may indicate a time-dependent decrease. Since only 6 hemp seed oil samples have been examined so far in 2021, it is impossible to make significant forecasts regarding whether the percentage of ARfD exceedance will decline in the next years.

In the case of the THC daily intake through the consumption of hemp tea samples, 7 of 10 (70%) samples exceeded the ARfD in 2016. In 2018, the lowest number of samples that exceeded the ARfD was observed (25%), followed by 61% in 2019. However, the percentage of samples with a daily intake of THC that exceeds the ARfD decreased for hemp tea from 2019 to 2021. For hemp tea samples, 11 of 18 (61%) in 2019, 14 of 34 (41%) in 2020, and 4 of 11 (36%) in 2021 exceeded ARfD due to their daily intake of THC.

In every year except 2016, the percentage of samples that exceeded the ARfD was higher for hemp food supplements than for hemp seed oil and hemp tea. In 2016, only 21% of the food supplement samples exceeded the ARfD, whereas 71% of the oil samples and 70% of the tea samples exceeded this toxicological assessment value. However, as shown in Figure 2, the mean daily intake of THC through the consumption of hemp food supplements (0.464 mg/d) was higher than that of hemp tea (0.050 mg/d) and hemp seed oil (0.366 mg/d) in 2016. This shows the toxicological relevance of some food supplements with a high THC content. However, the percentage of samples with a daily THC intake that exceeds ARfD decreased for hemp food supplements from 2018 to 2021. In the case of hemp food supplement samples, 7 of 9 (78%) in 2018, 23 of 31 (71%) in 2019, 34 of 62 (55%) in 2020, and 4 of 9 (44%) in 2021 exceeded the ARfD due to their daily intake of THC.

Unlike hemp seed oil and hemp tea, daily THC intake through the consumption of hemp food supplements exceeded the LOAEL in every year. Through the consumption of the hemp seed oil and hemp tea samples examined, the LOAEL was not exceeded in any year. In 2016, 2018, 2019, 2020, and 2021, 1 (7%), 4 (44%), 10 (32%), 7 (11%), and 1 (11%) hemp food supplement samples, respectively, exceeded the LOAEL. The LOAEL was exceeded by consumption of these 23 hemp food supplement samples, with an exhaustion of up to 674% (17 mg/d, 3400 mg/kg).

Furthermore, the daily THC intake of 10 of the 125 total hemp food supplement samples examined exceeded the minimum intoxication dose of 5 mg. The minimum intoxication dose was exceeded by 1 sample (7%) in 2016, by 2 samples in 2018 (22%), by 5 samples in 2019 (16%), and by 2 samples in 2020 (3%). The 2 highest THC contents in hemp food supplements were examined in 2019, where samples with a THC content of 2200 (daily intake 11 mg/d) and 3400 mg/kg (daily intake 17 mg/d) exhausted the minimum intoxication dose by 220% and 340%. This result shows that, also for the consumption of a dose of 5 g/d (i.e., less than one tablespoon), for 10 of 125 hemp food supplement samples, intoxication cannot be excluded. Due to their high mean THC content, some of the examined food supplements are particularly well suited to exceeding the minimum intoxication dose if they are used off label with a consumption quantity of more than 5 g/d.

In summary, in the years 2016, 2018, 2019, 2020, and the first half of 2021, daily THC intake of 38 of 103 hemp seed oil samples (37%), 38 of 81 hemp tea samples (47%), and 71 of 125 hemp food supplement samples (57%) exceeds ARfD. Furthermore, through the consumption of hemp food supplements, 23 of 125 samples (18%) exceed the LOAEL and 10 of 125 samples (8%) exceed the minimum intoxication dose of 5 mg. In conclusion, due to their higher mean THC content and their higher percentage of exceeded LOAEL and minimum intoxication doses, the hemp food supplements represent the highest hazard for the consumer regarding THC intake and they should be controlled with priority. The high THC content of hemp food supplements is confirmed by previous similar evaluations by BfR and EFSA, who determined that the THC content is much higher for hemp food supplements than for other hemp food products, such as beverages, oil, seeds, and tea [16,23].

## 4. Conclusions

The data evaluated in this study was made available as a response to a consumer inquiry, based on the Consumer Information Act, through the FragDenStaat.de website [19,20]. The anonymous requestor relates to the warnings of health consequences due to the psychotropic effects associated with THC in hemp food products. The requestor asked what THC content was examined in different hemp food product groups and what mean daily consumption can be used to estimate the risk to consumers regarding hemp food products.

For beverages, seeds, oil, and tea, the daily consumption of the NVS II was used while for food supplements, our own assumptions were used [10,25]. Under the usage of these daily consumptions, the mean daily THC intakes of 0.005 mg/d for beverages, 0.008 mg/d for seeds, 0.153 mg/d for oil, 0.157 mg/d for tea, and 1.360 mg/d for food supplements were calculated. Since the mean THC intake through the consumption of hemp beverages and hemp seeds does not exceed ARfD, the risk to the consumer is considered low, but cumulative effects should still be considered. Processing of the products by the consumer (e.g., roasting the seeds, adding seeds to baked goods, or using hemp seed oil in warm conditions) can cause the conversion from THCA to THC and thus affect the THC content. However, the mean THC intake of consumers of hemp seed oils, hemp tea, and hemp food supplements exceeds ARfD and exhausts LOAEL by 6% (oils), 6% (tea), and 54% (food supplements). Hemp food supplements, such as the currently popular CBD products, contain the highest THC content of all five examined product groups. Although the mean daily THC intake through the consumption of hemp food supplements does not exceed the LOAEL, some samples with an exceptionally high THC content were observed. For hemp food supplements, 23 of the 125 samples examined (18%) exceeded the LOAEL, while 10 of the 125 samples (8%) exceeded the minimum THC intoxication dose of 5 mg. The consumption of these food supplement samples can result in serious symptoms and risks for the consumer due to their high THC content [8,10]. However, a significant linear decrease in the THC content was observed in hemp food supplements between 2018 and the first half of 2021 (*n* = 111, R = −0.36, *p* < 0.0001). However, continuous control of hemp producers is still required to ensure low THC levels in hemp food products. Hopefully, the currently ongoing novel food approval procedure will lead to enforceable standards for CBD products. However, it would probably be more desirable to regulate lifestyle CBD products, especially those with questionable suitability as “food supplements”, outside the scope of foods inside a separate framework within the currently planned controlled distribution of cannabis to adults for recreational use in licensed stores in Germany.

## Figures and Tables

**Figure 1 foods-11-00486-f001:**
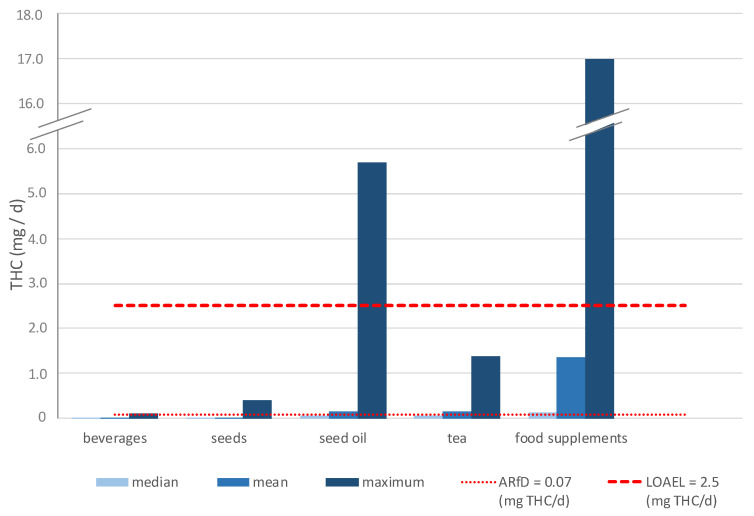
Median, mean, and maximum daily intake of ∆^9^-tetrahydrocannabinol (THC) from hemp tea, hemp seeds, hemp seed oil, food supplements, and beverages compared to the acute reference dose (ARfD) and lowest observed adverse effect level (LOAEL) for a person with a body weight of 70 kg, assuming the daily consumption shown in Table 2. Data from 2008–2021.

**Figure 2 foods-11-00486-f002:**
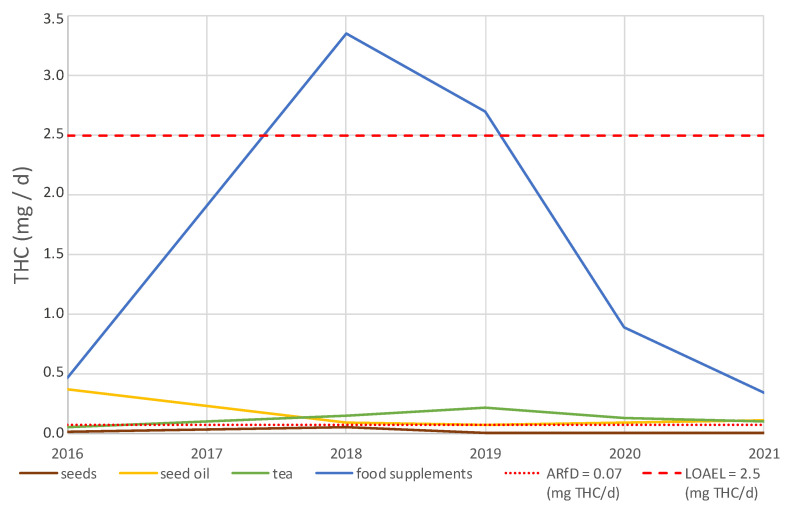
Mean daily intake of ∆^9^-tetrahydrocannabinol (THC) of hemp tea, hemp seeds, hemp seed oil, and hemp food supplements for 2016, 2018, 2019, 2020, and 2021 compared to the acute reference dose (ARfD) and lowest observed adverse effect level (LOAEL) for a person with a body weight of 70 kg, assuming the daily consumption shown in Table 2. Data for 2016, 2018, 2019, 2020, and the first half of 2021 were used.

**Figure 3 foods-11-00486-f003:**
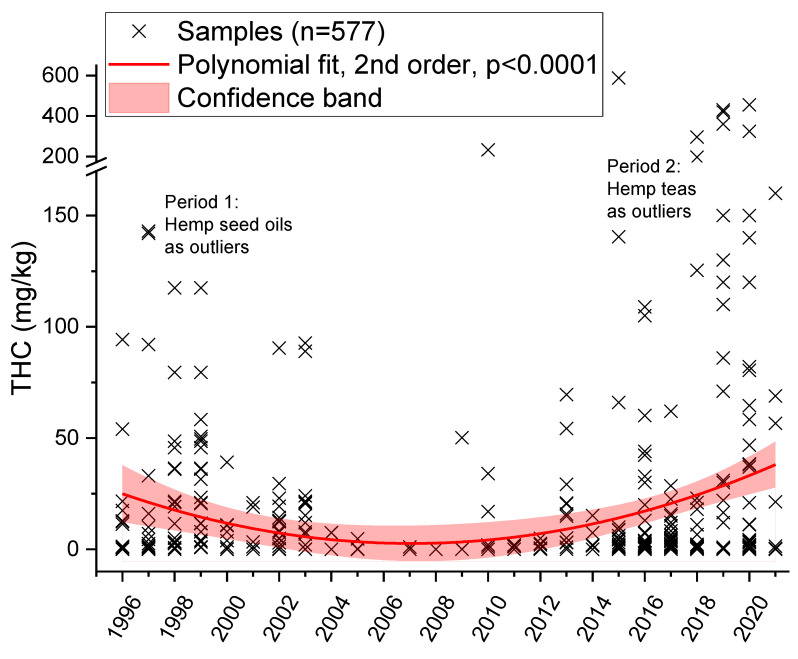
Time trend of the ∆^9^-tetrahydrocannabinol (THC) content in hemp food products, except hemp food supplements, in the years 1996 to the first half of 2021 (note: data before 2008 are total-THC).

**Figure 4 foods-11-00486-f004:**
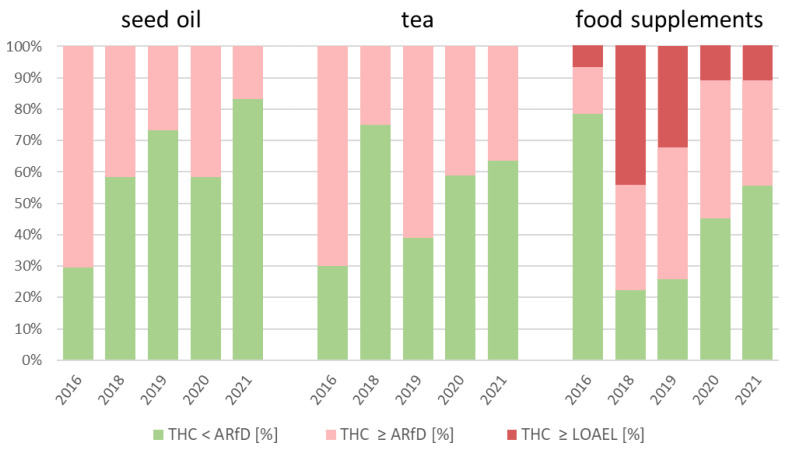
Percentage of the total number of samples with a daily intake of ∆^9^-tetrahydrocannabinol (THC) exceeding the acute reference dose (ARfD) and lowest observed adverse effect level (LOAEL) for a person with a body weight of 70 kg, assuming the daily consumption shown in Table 2 for hemp tea, hemp seed oil, and hemp food supplements.

**Table 1 foods-11-00486-t001:** Overview of the different products of the product groups hemp beverages, hemp seeds, hemp seed oils, hemp tea, and hemp food supplements.

Product Group	Product Categories Contained in the Product Group
beverages	ice tea, energy drinks, fitness drinks, fruit drinks, vegetable drinks, soft drinks
seeds	hemp seeds, grounded hemp seeds
seed oil	hemp seed oil
tea	hemp leave tea, hemp flower tea, herbal tea, unfermented tea, tea-like products, mixtures of tea-like products, flavored tea-like products
food supplements	food supplements including CBD oils, protein and amino acid supplements, vitamin supplements

**Table 2 foods-11-00486-t002:** Daily consumption, number of (quantifiable) samples, and maximum, mean, median, and 90th and 95th percentile of ∆^9^-tetrahydrocannabinol in hemp tea, hemp seed oil, hemp seeds, hemp food supplements, and hemp beverages.

Product Group	Mean Daily Consumption (g/d)	Number of Samples	Number of Quantifiable Samples	Maximum (mg/kg)	Mean (mg/kg)	Median (mg/kg)	90th Percentile (mg/kg)	95th Percentile (mg/kg)
beverages	176 [25]	61	19	0.53	0.03	0.001	0.05	0.10
seeds	2.0 [25]	93	89	196	4.1	0.4	2.7	4.0
seed oil	24.5 [25]	137	102	233	6.2	2.0	11.1	16.3
tea	2.3 [25]	89	78	588	67.3	20.9	152	346
food supplements	5 [10]	131	98	3400	272	26.0	720	1265

**Table 3 foods-11-00486-t003:** Number of samples of hemp tea, hemp seed oil, hemp seeds, and hemp food supplements for 2016, 2018, 2019, 2020, and 2021 ^1^.

Product Group	2016	2018	2019	2020	2021 ^1^
tea	10	8	18	34	11
seeds	36	8	9	14	8
seed oil	17	12	56	12	6
food supplements	14	9	31	62	9

^1^ Data for 2021 were only partially available.

## Data Availability

Publicly available datasets were analyzed in this study. These data can be found here: https://fragdenstaat.de/anfrage/thc-in-lebensmitteln/#nachricht-624273 (accessed on 6 December 2021). The derivative calculations presented in this study are available in the Appendix A.

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
