# Peer review of "Time Trends of Tetrahydrocannabinol (THC) in a 2008–2021 German National Survey of Hemp Food Products"

_foods, 2022, doi:10.3390/foods11030486_

Round 1

Reviewer 1 Report

The authors describe time trends of THC contents in certain product groups containing hemp in Germany. The paper has its importance in demonstrating problematic areas in food (supplement) trends. 

In the plant, the native form of the cannabinoid discussed here is non-psychoactive THCA which is partly already converted to the psychoactive THC in the plant. However, THCA is easily converted into THC (e.g. by heat). Therefore it is in many regulations forbidden as well and many regulations limit the sum of THC and THCA and not neutral THC only.   

Part of the analysis were performed with GC/MS. In gascrhomatography, the hot temperature in the injector is sufficient to convert almost all THCA into THC, leading to overestimates of neutral THC compared to LC methods. Using the sum of THCA + THC, no overestimation would occur and results from both methods - GC and LC - can be used without discrimination. Considering the time series, shifting from GC to LC in routine analysis may have had an impact on results.

Ideally, reporting both - THC and (THCA+THC) would be the best option. However, that would also increase the extent of this paper too much. 

Processing of food products containing hemp is crucial for the conversion of THCA into THC. Processing does not only include processing until the finished product is sold on the market (and analysed by food control), but also processing by the consumer. Seeds are e.g. roasted or added to baked goods. Seed oil cannot be used for high temperature cooking. However, it can be heated up to 160°C which will again convert much of THCA into THC.  That could be addressed a bit more in the specific product groups and in risk assessment. 

Minor issues:

  • 'Cannabis sativa' in the keywords should be in italics.
  • Figure 2: check y-axis.

Author Response

The authors describe time trends of THC contents in certain product groups containing hemp in Germany. The paper has its importance in demonstrating problematic areas in food (supplement) trends. 

In the plant, the native form of the cannabinoid discussed here is non-psychoactive THCA which is partly already converted to the psychoactive THC in the plant. However, THCA is easily converted into THC (e.g. by heat). Therefore it is in many regulations forbidden as well and many regulations limit the sum of THC and THCA and not neutral THC only.   

Part of the analysis were performed with GC/MS. In gaschromatography, the hot temperature in the injector is sufficient to convert almost all THCA into THC, leading to overestimates of neutral THC compared to LC methods. Using the sum of THCA + THC, no overestimation would occur and results from both methods - GC and LC - can be used without discrimination. Considering the time series, shifting from GC to LC in routine analysis may have had an impact on results.

Ideally, reporting both - THC and (THCA+THC) would be the best option. However, that would also increase the extent of this paper too much. 

RESPONSE: We are aware of the analytical difficulties that come with the selective determination of neutral THC and the transition from GC-data to LC-data. As the reviewer correctly mentioned, reporting both (THC and the sum of THC and THCA) would be the best option but would extend this paper too much. Considering the pro and cons, we decided to use only the data of neutral THC contents for our evaluation (except for Figure 3). Our main reasons for the decisions were:

The THC contents were determined by German state laboratories, which are required to have a quality management system, certifications and validation of methods. Hence, we assume that when the laboratories did not apply LC, a correction system for GC analytics was implemented to make sure that the values of single determination of neutral THC are correct (e.g. using derivatization).

Furthermore, the number of samples for the different data had an impact on our decision. For neutral THC, the data contain 511 samples, for the sum of THC and THCA only 289. By choosing to use only the data of neutral THC for our evaluation, the evaluation is much more representative due to the higher number of samples.

The most important hemp-related toxicological assessment values of the EU, the acute reference dose (ARfD) and the lowest observed adverse effect level (LOAEL), are only applicable to the neutral THC content. Neutral THC is psychotropic, whereas THCA is nonpsychotropic. Nowadays, there are no toxicological assessment values for the sum of THC and THCA. In 2000, the predecessor of the German Federal Institute for Risk Assessment (BfR) derived guideline values for the sum of THC and THCA in food [1]. Since EFSA publicly released the new toxicological assessment values (ARfD and LOAEL) in 2015 for the EU, the guideline values are considered outdated and no longer up-to-date. In this paper, we want to estimate the toxicological risk for consumers of hemp food products. Therefore, we have to compare the contents of the hemp food products with toxicological assessment values. Because the scientific up-to-date toxicological assessment values (ARfD and LOAEL) only account for the neutral THC content, the evaluation of the neutral THC content makes more sense than the evaluation of the sum of THC and THCA.

[1] BgVV Pressedienst (2000). BgVV empfiehlt Richtwerte für THC (Tetrahydrocannabinol) in hanfhaltigen Lebensmitteln. https://www.bfr.bund.de/de/presseinformation/2000/07/bgvv_empfiehlt_richtwerte_fuer_thc__tetrahydrocannabinol__in_hanfhaltigen_lebensmitteln-884.html (acessed on 18.01.2022).

Processing of food products containing hemp is crucial for the conversion of THCA into THC. Processing does not only include processing until the finished product is sold on the market (and analysed by food control), but also processing by the consumer. Seeds are e.g. roasted or added to baked goods. Seed oil cannot be used for high temperature cooking. However, it can be heated up to 160°C which will again convert much of THCA into THC.  That could be addressed a bit more in the specific product groups and in risk assessment. 

RESPONSE: The mentioned aspect has been included in the text.

Minor issues:

  • 'Cannabis sativa' in the keywords should be in italics. RESPONSE: The changes have been incorporated into the text.
  • Figure 2: check y-axis. RESPONSE: Figure 2 has been checked and the change has been incorporated into the text.

Reviewer 2 Report

The following suggestions can be considered.

Cannabinoids are the most important chemical compounds found in the Cannabis sativa species, including Δ9-tetrahydrocannabinol (Δ9-THC), cannabidiol (CBD), cannabinol (CBN), cannabigerol (CBG), and cannabichromene (CBC). Why only Tetrahydrocannabinol (THC) have been evaluated?

-The method of THC extraction must be clearly described.

-The specifications of the devices and methods used for chromatography (GC and HPLC) should be explained.

 -What statistical design and software has been used to analyze the data?Must be explained.

-The nature of the research material must be specified. Fresh matter or dry matter or extract? The units in the charts must be completed.

Author Response

Cannabinoids are the most important chemical compounds found in the Cannabis sativa species, including Δ9-tetrahydrocannabinol (Δ9-THC), cannabidiol (CBD), cannabinol (CBN), cannabigerol (CBG), and cannabichromene (CBC). Why only Tetrahydrocannabinol (THC) have been evaluated?

RESPONSE: Our work is based on a consumer’s inquiry into the THC content in food products and the expected exposures resulting. THC is the most abundant psychotropic substance in the hemp plant. The cannabinoids CBD, CBN, CBG, and CBC, on the other hand, are not considered psychotropic or are psychotropic to a much lesser degree. With regard to food regulatory aspects and the food safety assessment the THC is the most important component, which is why we have focused on THC. In our opinion, the consideration of other cannabinoids would lead to far in this publication.

 -The method of THC extraction must be clearly described.

-The specifications of the devices and methods used for chromatography (GC and HPLC) should be explained.

RESPONSE: Both the extraction methods as well as the specifications of the devices and methods used for chromatography (GC and HPLC) were not a part of the data. Therefore, we had no information about these aspects.

Our data evaluation and risk assessment on this topic are based on a data set from the German Federal Office of Consumer Protection and Food Safety (BVL). The BVL is a German authority for consumer protection and they collect in the context of a monitoring investigation results on food safety relevant topics e.g. THC in food. This data collection is in cooperation with federal state laboratories. Since the laboratories of the federal states are accredited laboratories, we assume that all analyses are performed with validated methods that meet the criteria according to ISO/IEC 17025.

 -What statistical design and software has been used to analyze the data? Must be explained.

RESPONSE: The explanation of the statistical design has been verified. In our opinion, all statistical calculations, like the calculation of mean, median, and maximum, were described properly. The used software has been incorporated into the text.

-The nature of the research material must be specified. Fresh matter or dry matter or extract? The units in the charts must be completed.

RESPONSE: All contents given in the text and the figures refer to the consumption quantities described in the method section. The method section also describes the type of material (fresh matter, dry matter, or extract) to which the consumption quantity refers.

The units in the figures have been checked.